# A Multistep Interval Prediction Method Combining Environmental Variables and Attention Mechanism for Egg Production Rate

Hang Yin [1,2,*], Zeyu Wu [2], Jun-Chao Wu [3], Yalin Chen [2], Mingxuan Chen [2], Shixuan Luo [2], Lijun Gao [4] and Shahbaz Gul Hassan [2,*]

1   College of Big Data and Internet, Shenzhen Technology University, Shenzhen 518118, China
2   College of Information Science and Technology, Zhongkai University of Agriculture and Engineering, Guangzhou 510225, China; wuzeyu0929@163.com (Z.W.); 18148881808@163.com (Y.C.); a13662513987@126.com (M.C.); sx1819395734@163.com (S.L.)
3   Institute of Collaborative Innovation, University of Macau, Macao 999078, China; junchao.wu@connect.um.edu.mo
4   College of Computer Science, Shenyang Aerospace University, Shenyang 110136, China; gaolijun@sau.edu.cn
*   Correspondence: yinhang@sau.edu.cn (H.Y.); mhasan387@zkhu.edu.cn (S.G.H.)

**Abstract:** The egg production rate is a crucial metric in animal breeding, subject to biological and environmental influences and exhibits characteristics of small sample sizes and non-linearity. Currently, egg production rate prediction research predominantly focuses on single-step point prediction, lacking multistep and interval prediction exploration. To bridge these gaps, this study proposes a recursive, multistep interval prediction method for egg production rates, integrating environmental variables and attention mechanisms. Initially, this study employed three gradient boosting tree models (XGBoost, LightGBM, CatBoost) and the recursive feature elimination (RFE) method to select critical environmental variables and reduce data dimensionality. Subsequently, by scaling the time scale of important environmental variables and utilizing the variational modal decomposition improved by the grey wolf optimization (GWO-VMD) method for time-series decomposition, the volume of important environmental variable data is augmented and its complexity is reduced. Applying the long short-term memory (LSTM) neural network to obtain direct multistep predictions on IMFs, the predicted outcomes are averaged daily to yield the environmental variables for the upcoming two days. Finally, a multistep interval prediction model based on Seq2seq-Attention and Gaussian distribution is proposed in this study, and parameter optimization is carried out using the multi-objective grey wolf optimization algorithm (MOGWO). By inputting the historical egg production rate data and environmental variables into the proposed model, it is possible to achieve multistep point and interval prediction of egg production rates. This method was applied to analyze a dataset of egg production rates of waterfowl. The study demonstrated the feasibility of the recursive multistep prediction approach combined with environmental variables and guides egg production estimation and environmental regulation in animal husbandry.

**Keywords:** egg production rate; multistep point prediction; interval prediction; environmental variables selection; GWO-VMD decomposition; Seq2seq-Attention; MOGWO optimize parameter





## 1. Introduction

Animal husbandry is a crucial component of agriculture [1]. Nowadays, animal husbandry is gradually moving towards intelligentization. IoT and AI technologies used to address practical problems are the leading research directions in this field [2]. Predicting the egg production rate can be utilized for measuring the success of the animal population, predicting egg yields, and optimizing future income [3,4]. Therefore, predicting future egg production rates is significant to the animal husbandry industry.

From an environmental perspective, the environmental factors that affect egg production rates include temperature, hydrogen sulfide, light, ammonia, carbon dioxide, humidity, dust, etc. Kim et al. [5] investigated how various temperatures and humidity affected laying chickens' ability to produce eggs. Geng et al. [6] investigated the effects of separate and combined lighting and photoperiod on egg production and quality. In [7,8], Shepherd and Saksrithai pointed out that poultry houses are prone to producing ammonia, dust, greenhouse gas and hydrogen sulfide, and that this can have a negative impact on how well animals and fowl produce eggs. Therefore, this study mainly analyzes the impact of different environmental variables on egg production rates from an environmental perspective. It combines multiple environmental variables to predict the egg production rate.

Statistical analysis, fuzzy modeling, machine learning (ML), and deep learning (DL) models are currently the most widely used techniques for estimating egg production rates. Abdallah et al. [9] considered egg production data as a time series and utilized the ARIMA statistical model to forecast egg production. An optimal fuzzy prediction model for egg production was proposed by Omomule et al. [10] using young age, feed quantity and quality, chicken weight, and total egg production as data points. With the rise of artificial intelligence (AI), machine learning (ML) models have been applied to the field of egg production rate prediction. Minlan et al. [11] used principal component analysis (PCA) to aggregate the effects of various factors influencing egg production rates. They then employed particle swarm optimization-least squares support vector machines (PSO-LSSVM) to construct a weighted data regression model, obtaining better predictions than conventional models. Gonzalez-Mora et al. [12] utilized random forest (RF) to predict egg production by combining indoor environmental and hygrothermal variables. In addition, the feature importance ranking function of RF allows for scenario analysis and is a good framework for evaluation. Recently, deep learning (DL) models have become known for their excellent problem-solving capabilities and have been used in complex and non-linear egg production prediction problems. Ghazanfari et al. [13] employed an artificial neural network (ANN) with two hidden layers to successfully learn the relationship between the hen's age and egg production. Felipe et al. [14] utilized multiple linear regression (MLR), Bayesian networks (BN), and artificial neural networks (ANN) to forecast the total egg production (TEP) of European quails. Their study revealed the presence of non-linear relationships between different variables and egg production and selecting appropriate covariates enhanced the accuracy of egg production prediction. Liu et al. [15] employed gray relational analysis to examine the correlation between multiple environmental variables, feed intake, and the egg production rate. They established deep belief networks with swarm optimization algorithms (PSO-DBN), which outperformed other existing combined models to forecast the egg production rate.

There has been some progress in research on the methods for predicting egg production rates. However, there are still some drawbacks that need to be addressed, which are as follows:

(1)　Currently, the methods used to reduce the factors that reduce egg production rates primarily involve statistical analysis or feature extraction, but these approaches suffer from issues such as inadequate feature selection or low interpretability of extracted features.

(2)　Research on egg production rate prediction mainly focuses on single-step prediction, lacking studies on multistep prediction. Multistep prediction can provide more helpful information, which is significant for production estimation and regulation.

(3)　Research on egg production rate prediction has focused solely on point prediction, while lacking studies on interval prediction. Interval prediction can quantify the unavoidable bias brought about by multistep point prediction and better describe the uncertain information about egg production rate.

Feature selection reduces data dimensionality, training time and memory, while also mitigating the effects of noise and extraneous variables, improving model performance [16]. Based on the function of feature importance calculation, gradient boosting decision trees

(GBDT) using decision trees as weak learners have become a novel and effective feature selection model, while XGBoost, LightGBM and CatBoost are three new models that improve GBDT. They exhibit fast training speeds and little dependency on parameters [17]. Chen and Zhang [18,19] used XGBoost in conjunction with the RFE approach for feature selection, demonstrating that this method produces better dimensionality reduction and is more efficient than other feature extraction methods. In addition, in [20,21], Banga and Karbasi employed LightGBM and CatBoost as feature selection models, respectively, and similarly achieved excellent feature selection results. However, the efficacy of XGBoost, LightGBM, and CatBoost models depends on the data size and model parameters [22], necessitating careful analysis and comparison based on specific circumstances.

Environmental variables are suitable for direct prediction because of their short time scale and large amount of data compared with the egg production rate. To achieve recursive multistep prediction of the egg production rate, we need to first perform multiple direct-step predictions of the screened important environmental variables, and then use future environmental variables as input for recursive multistep prediction. Kim et al. [23] used the LSTM neural network to achieve multistep prediction of environmental variables by setting appropriate input and output time windows. However, environmental variables are volatile and complex. To further improve the multistep prediction performance, preprocessing of environmental variables is necessary to reduce complexity. Time–frequency domain decomposition is a useful method for time-series decomposition and noise reduction. Among the common decomposition methods, variational modal decomposition (VMD) has better decomposition performance and is less affected by noise [24]. The total number of decomposition modes (K) and the quadratic penalty coefficient ($\alpha$) have a significant impact on the decomposition results of the actual signal during the VMD decomposition process and are difficult to determine. In the field of fault diagnosis, the variational modal decomposition improved by the grey wolf optimization (GWO-VMD) method was proposed to solve the problem of determining VMD parameters [25]. Multiple IMFs with low complexity and high regularity can be obtained by optimizing parameters K and $\alpha$ with the local minimal envelope entropy as the fitness function using the grey wolf optimization algorithm.

Interval prediction can effectively evaluate the risks caused by point prediction errors and provide uncertainty information [26]. In [27,28], Huang and Abbaszadeh performed interval predictions for dissolved oxygen and crop yield, respectively, to obtain more accurate and reliable predictions. Seq2seq-Attention has been widely applied in various fields for time-series forecasting [29,30]. It can extract valid information from multidimensional and historical data more efficiently, and thoroughly explore the cause-and-effect relationship between each variable and the target [31]. However, there are fewer applications in agriculture. This study proposes a new probabilistic recurrent neural network based on Seq2seq-Attention and Gaussian distribution to capture the correlation between the egg production rate and multiple environmental variables, and accomplish multistep point and interval prediction of egg production.

Intelligent optimization algorithms have been widely used in the process of determining neural network initialization parameters to solve the problem of the difficult determination of parameters [11,15]. This study adopted the multi-objective grey wolf optimization algorithm (MOGWO), which is inspired by the hunting behavior of wolves in a pack to search for the optimal initial parameter combination of the model. Two objectives were defined for optimization, resulting in higher prediction accuracy and stability than a single objective.

Based on the above, this study proposes a hybrid egg production rate prediction method combining environmental variables and attention mechanisms to obtain accurate point prediction and interval prediction results. The main contributions and innovations of this study are as follows:

- A feature selection method is proposed in this study that combines XGBoost, Light-GBM, CatBoost, and the RFE feature elimination approach. This method can filter

out redundant environmental variables more thoroughly, ultimately reducing model training time and improving prediction accuracy.

- A multistep prediction strategy for egg production rates is also proposed. Due to limited egg production rate data, direct multistep prediction yields unsatisfactory results. Based on the idea of recursive multistep forecasting, we first performed autoregressive multistep forecasting of environmental variables on small time scales, and then averaged the forecasting results daily and combined them with historical egg production rate data to achieve multistep forecasting of future egg production rates.

- Furthermore, an egg production rate interval prediction model is introduced in this study. Compared to the current egg production rate prediction models, this model incorporates seq2seq architecture and attention mechanisms to enhance the utilization of environmental and egg production rate data and improve point prediction accuracy. Moreover, this model can output suitable prediction intervals to measure the uncertainty of the egg production rate. Lastly, the MOGWO multi-objective algorithm is used for optimization to ensure the accuracy and stability of both point and interval predictions.

## 2. Materials and Methods

The method's architecture can be divided into the following four modules: data preprocessing and organization, feature selection, environmental variable prediction, and egg production rate prediction. The first module focuses on preprocessing the environmental data collected from sensors and averaging it over 6 h and 24 h to obtain two types of data for multistep prediction of environmental variables and important feature selection. The second module uses XGBoost, LightGBM, and CatBoost models in combination with the RFE method to perform feature selection. The environmental variables are first sorted by importance based on the tree's importance calculation function. The three models are used together with historical egg production rate data and environmental variables to find the optimal number of features using RFE. The third module consists of the environmental variable decomposition module and the autoregressive multistep prediction module. The GWO-VMD algorithm decomposes complex environmental variables into multiple simple and easily predictable IMFs. The decomposed IMFs are then passed into the LSTM neural network separately for direct multistep prediction. Finally, the predicted results are averaged daily to obtain the future two-day environmental variables. The fourth module predicts egg production rates by using environmental variables and historical data. It utilizes an egg production rate interval prediction model proposed in this study to make both point and interval predictions for the next three days. The system architecture is shown in Figure 1.

### 2.1. Study Area and Data Source

This research used data collected at the Zhongcun Chinese Goose Breeding Base in the Panyu District of Guangzhou City, Guangdong Province. As shown in Figure 2a, a remote monitoring platform for waterfowl intensive breeding based on the Internet of Things has been developed. To carry out this experiment, we installed several Internet of Things (IoT) devices at the goose house to monitor environmental factors such as temperature, dust, carbon dioxide, light, humidity, ammonia, and hydrogen sulfide, as shown in Figure 2b.

The environmental data collected by IoT sensors will be transmitted to a remote cloud service center via a gateway and then stored in a database, making it easy for users to view and save the data on their desktop computers and mobile devices.

To ensure consistency of the samples from different seasons and periods, the study adopted environmental monitoring equipment provided by Guangzhou Hairui Information Technology Co., Ltd. (Guangzhou, China). The equipment includes a network transmission system, hub, temperature sensor, humidity sensor, $CO_2$ concentration sensor, light sensor, ammonia concentration sensor, noise sensor, total suspended particle sensor and $H_2S$ sensor. The equipment has a response time of less than or equal to 30 s, repeatability of less than or

equal to ±2%, linear error of less than or equal to ±2%, and zero drift of less than or equal to ±1%. The specific parameters are listed in Table 1.

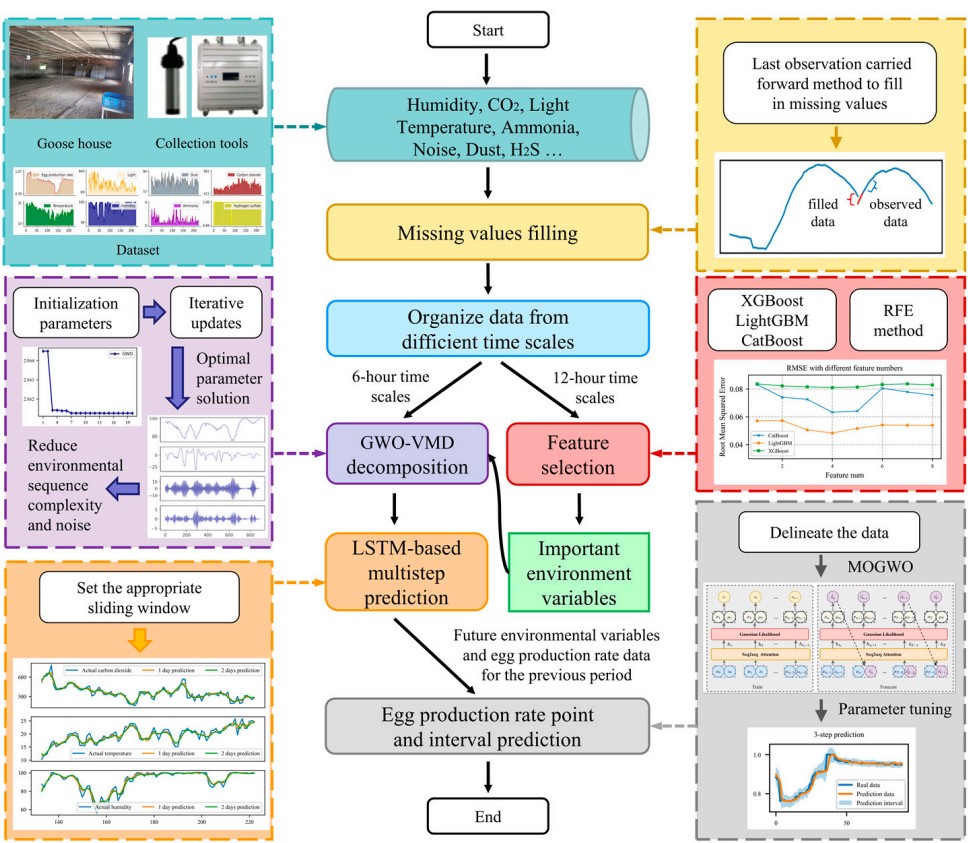

**Figure 1.** The overall process of predicting egg production rate for waterfowl.

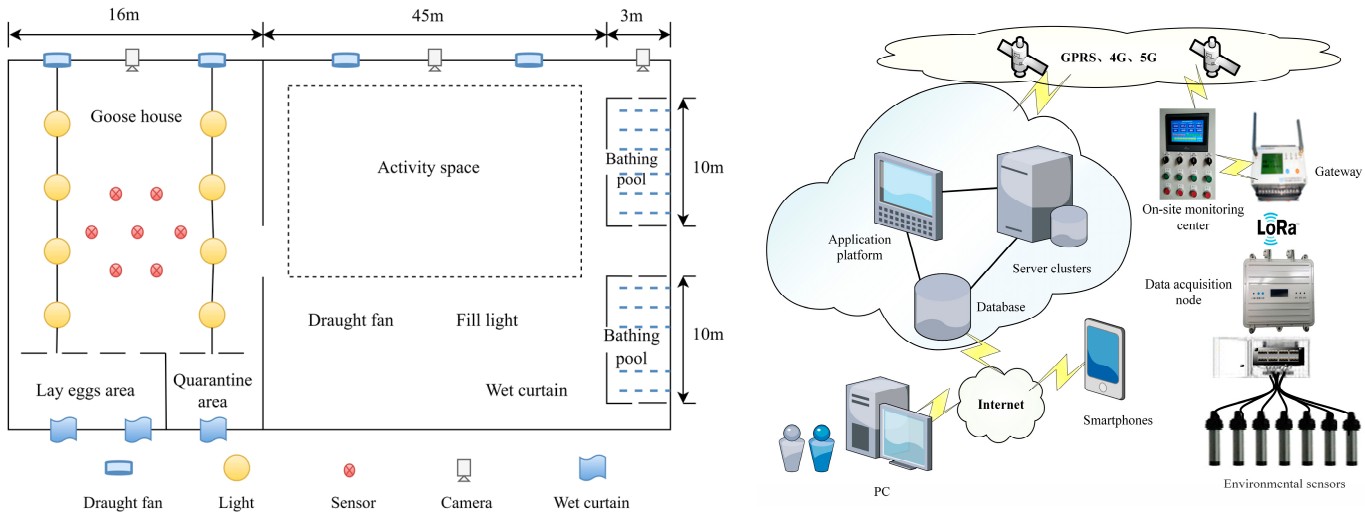

(**a**) Intensive farming environment planar diagram.

(**b**) Network topology diagram of IoT monitoring.

**Figure 2.** IoT-based remote monitoring platform of intensive aquaculture environment for Chinese goose.

**Table 1.** Technical data of sensors.

| Environmental Variables | Measurement Range | Precision | Agreement |
|---|---|---|---|
| Carbon dioxide (ppm) | 0~50,000 | ±20 | PWM |
| Temperature (°C) | −40~105 | ±0.4 | IIC |
| Humidity (%) | 0~100 | ±5 | IIC |
| Dust (ppm) | 0~999.9 | ±7% | Modbus |
| Ammonia (ppm) | 0~100 | ±5% | Modbus |
| Light (lx) | 0~65,535 | ±5 | IIC |
| Hydrogen sulfide (ppm) | 0~100 | ±3% | PWM |

*2.2. Feature Selection Method*

2.2.1. Gradient Boosting Tree Model

The gradient boosting tree is an indispensable application of the gradient boosting technique, which mainly utilizes base learners to solve problems in multiple stages while effectively preventing overfitting by optimizing the loss function. XGBoost [32] is a powerful, flexible, and portable tool that effectively leverages resources and overcomes the previous limitations of gradient boosting. It incorporates a regularization term in the cost function to control model complexity and prevent overfitting. Additionally, XGBoost optimizes the loss function using second-order Taylor expansion, thereby accelerating the optimization process. To reduce training time, Ke et al. [33] developed LightGBM, which utilizes an improved histogram algorithm to remove feature values before training, reducing traversal time. Furthermore, LightGBM utilizes the Exclusive Feature Bundling (EFB) algorithm to bundle mutually exclusive features to reduce the feature number and employs the Gradient-based One-Side Sampling (GOSS) algorithm to retain large-gradient samples while focusing on small-gradient samples. CatBoost [34] uses fully symmetric trees as base models to compute leaf values, thereby addressing gradient bias, prediction shift and overfitting issues. Moreover, CatBoost incorporates an automatic transformation algorithm to convert categorical features into numerical features, facilitating the efficient processing of categorical features and overall improving the model accuracy and generalization ability.

2.2.2. Filtering Methods for Important Environment Variables

Recursive Feature Elimination (RFE) is a wrapper method for feature selection that iteratively eliminates the least essential features using a specific learning algorithm. This experiment uses XGBoost, LightGBM, and CatBoost as external learning algorithms for feature selection. By comparing and contrasting the outcomes of the three models' feature selection, it is intended to determine the optimum number of features for the input model. The primary process involves sorting the feature importance of each subset of features in each round based on the external learning algorithm results, filtering out low-importance features to reduce the feature dimensionality, and continuously updating feature importance. The feature subset with the maximum predictive accuracy is determined by integrating the various feature subsets and the results of the external learning algorithm.

*2.3. Optimization Algorithm Based on Grey Wolf Pack*

2.3.1. Grey Wolf Optimizer Algorithm

Inspired by the foraging behavior of wolf packs, Mirjalili et al. [35] proposed a single-objective swarm intelligent optimization algorithm called the grey wolf optimizer (GWO) algorithm. A grey wolf pack consists of four types of wolves, namely α, β, δ, and ω, which can be classified from high to low according to their rank. The α, β, and δ wolves lead the pack's hunting behavior, while the remaining ω wolves follow their lead. The wolf pack approaches the optimal solution in the search space through the initial solutions of the α, β, and δ wolves. The optimal solution is obtained by continuously updating the position of the wolves and narrowing the distance to the prey.

### 2.3.2. Multi-Objective Grey Wolf Optimization Algorithm

Mirjalili et al. [36] improved the single-objective grey wolf optimizer algorithm and proposed a multi-objective grey wolf optimizer (MOGWO) algorithm that is suitable for solving multi-objective problems. Compared with GWO, MOGWO introduces an external population archive to store current non-dominated Pareto optimal solutions and changes the leader selection strategy. This algorithm inherits the advantages of GWO, such as fewer parameters, ease of implementation, and fast convergence rate.

### *2.4. GWO-VMD Method*

### 2.4.1. Variational Mode Decomposition

Variational mode decomposition (VMD) [37] is an adaptive and fully non-recursive signal processing method that effectively addresses endpoint effects and mode mixing issues in decomposing raw signals that arise in EMD. It demonstrates excellent performance in decomposing non-linear and complex signals. By pre-determining the number of decomposed modes, VMD can adaptively match each mode's optimal central frequency and limited bandwidth, effectively separating the intrinsic mode functions (IMF) and achieving signal frequency division. Finally, it obtains the practical decomposition components of the given signal. The original signal can be decomposed into K intrinsic mode functions (IMFs) by VMD decomposition, and K is the number of decomposed modes that need to be predetermined.

### 2.4.2. Envelope Entropy

Envelope entropy is an excellent indicator of the sparsity of each IMF obtained from the decomposition. When there is more noise in the IMF, the envelope entropy is more prominent, and vice versa. Its fundamental concept is to extract the signal's characteristics by combining Hilbert transform with information entropy. The mathematical formulas are as follows:

$$E_p = -\sum_{j=1}^{m} p_j \lg p_j \tag{1}$$

$$p_j = a(j) / \sum_{j=1}^{m} a(j) \tag{2}$$

Where $E_p$ denotes the envelope entropy; $a(j)$ represents the envelope signal sequence obtained by Hilbert demodulation of the signal $x(j)$ ($j = 1, 2, \ldots, m$); $p_j$ is the normalized form of $a(j)$.

### 2.4.3. Environmental Time-Series Decomposition Method Based on GWO-VMD

In decomposing environmental time series, the total number of decomposition modes (K) and the quadratic penalty coefficient ($\alpha$) have a significant impact on the decomposition performance of VMD. To determine the VMD parameters, this study used variational modal decomposition improved by the grey wolf optimization (GWO-VMD) method, which uses the local minimum envelope entropy as the fitness function and employs the GWO algorithm to optimize the parameters K and $\alpha$. The envelope entropy can reflect the sparsity of the IMFs obtained by decomposing the original sequence. The smaller the envelope entropy, the less noise the IMF contains. Therefore, the more stable and favorable it is for prediction. The main steps of GWO-VMD are as follows:

Step 1. Initialize the grey wolf population size N and set the maximum iteration of the algorithm to M.

Step 2. Define the optimization ranges for the VMD parameters K and $\alpha$, and initialize the other VMD parameters.

Step 3. Iteratively update the grey wolf population information to find the optimal parameter combination.

Step 4. Apply the optimized parameter combination (K, α) to VMD for decomposition and calculate the value of the fitness function pair.

Step 5. Check if the iteration conditions are met and proceed to step 6 if satisfied, or return to step 3 otherwise.

Step 6. Stop updating the iterations and obtain the optimized combination of parameters (K, α).

*2.5. New Interval Prediction Model for Egg Production Rate*

2.5.1. Long Short-Term Memory

The LSTM network [38] is neural network based on the improvement of the RNN. It aims to introduce a cell state that stores information and three additional gates (input gate, forget gate, and output gate) that control the amount of information flow to solve the long-term dependency problem of RNN. Figure 3 illustrates the internal structure of an LSTM unit. The forget gate $f_t$ determines how much information from the previous cell state $c_{t-1}$ should be retained in the current cell state $c_t$, while the input gate $i_t$ decides how much information to include in the current cell state $c_t$. The output gate $o_t$ controls how much information of the current cell state $c_t$ is retained in the output $h_t$ at the current time.

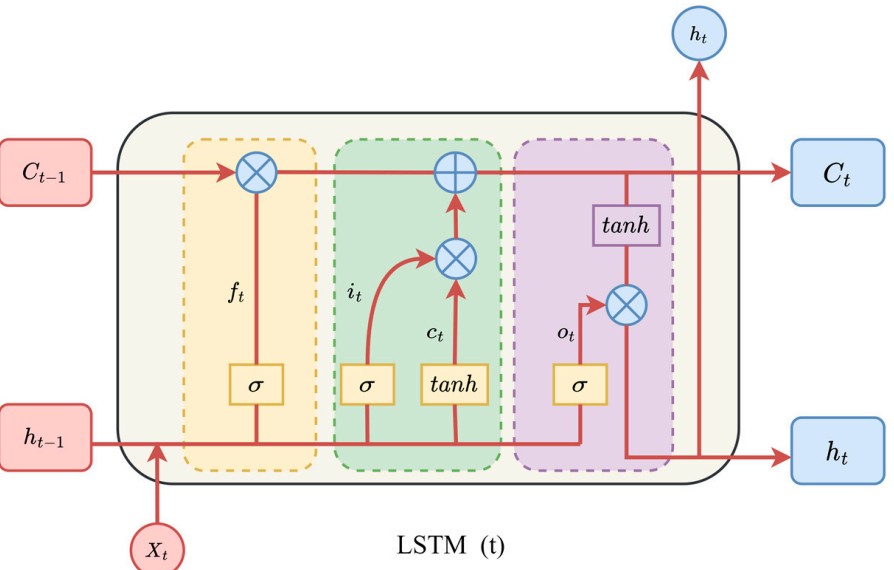

**Figure 3.** LSTM architecture.

2.5.2. Seq2seq-Attention Architecture

The attention mechanism is inspired by how the human brain selectively focuses on important information while ignoring irrelevant details, thereby improving the performance and efficiency of the model. In the field of time-series forecasting, traditional encoder–decoder architecture is used. It encodes the input time series into a fixed-length vector, no matter its length. This may overlook important temporal information and affect the accuracy of prediction, especially for multidimensional and multivariable data. By contrast, the attention mechanism can capture the temporal correlation between multidimensional time series and extract the relationship between input and output features, thus improving prediction accuracy. Figure 4 illustrates the structure of incorporating the soft attention mechanism into the seq2seq framework.

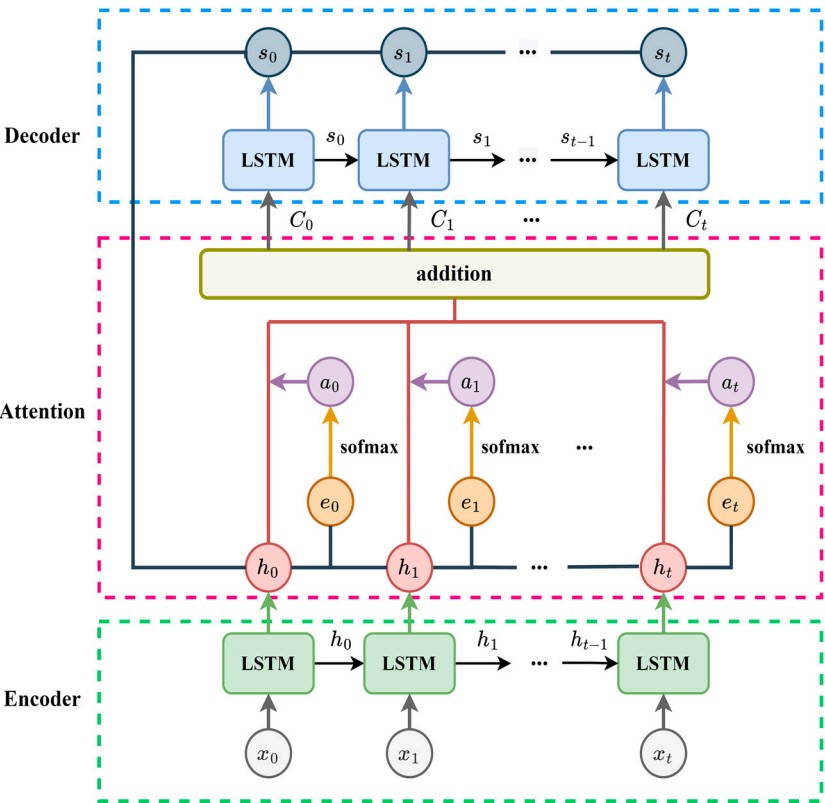

**Figure 4.** Seq2seq-Attention architecture.

By considering the relationship between the context vector and the encoder's hidden layer output $h_\beta$, the attention mechanism recreates a new context vector C, which is then passed into the decoder. The context vector $C_i$ is calculated by the attention weight $a_{i,j}$ and the hidden layer output $h_\beta$. The formulas are as follows:

$$C_i = \sum_{j=1}^{n} a_{i,j} h_\beta \tag{3}$$

$$a_{i,j} = \frac{\exp(e_{i,j})}{\sum_{k=1}^{T} \exp(e_{i,k})} \tag{4}$$

where represents the matching score between the input surrounding the position of j and the output surrounding the position of i. The scoring model comprises a single hidden layer neural network that is jointly trained with the other parts of the model. The non-linear activation function tanh is employed. The formula for $e_{i,j}$ is as follows:

$$e_{i,j} = V_a^T \tanh(W_a[S_{i-1}, h_\beta]) \tag{5}$$

where $V_a$ and $W_a$ represent weight matrices that need to be learned in the model, while $S_{i-1}$ denotes the hidden state of the decoder.

### 2.5.3. Neural Networks Based on Gaussian and Seq2seq-Attention

This study proposes a supervised learning model that can perform interval prediction using time-series data to achieve probability prediction of egg production rates. The structure is shown in Figure 5; assuming $z_t$ is the value of the time series at time t, and the current egg production rate data are $[z_0, z_1, \ldots, z_{t_0-2}, z_{t_0-1}]$, the model's objective is to predict the probability distribution p of the data $[z_{t_0}, z_{t_0+1}, \ldots, z_T]$ for the subsequent T time steps, where environmental factors are represented by the data $[x_0, x_1, \ldots, x_{t_0}, \ldots, x_T]$.

The time interval $[0 : t_0 - 1]$ is the training time scale, and $[t_0 : T]$ is the prediction time scale. The model learns from the data within the training time scale to predict the values within the prediction time scale. With these conditions, we can construct the probability distribution p as follows:

$$p_\Theta\left(z_{t_0:T}\middle|z_{0:t_0-1}, x_{1:T}\right) = \prod_{t=t_0}^{T} p_\Theta(z_t|z_{0:t-1}, x_{0:T}) = \prod_{t=t_0}^{T} \ell\left(z_t|\theta(h_t, \Theta)\right) \tag{6}$$

$$h_t = h(h_{t-1}, z_{t-1}, x_t, \Theta) \tag{7}$$

where h refers to the Seq2seq-Attention network; $h_t$ denotes the neural network output at a specific time. $\Theta$ represents the model parameters, while $\ell$ represents the likelihood function, with $\theta$ being the likelihood parameter.

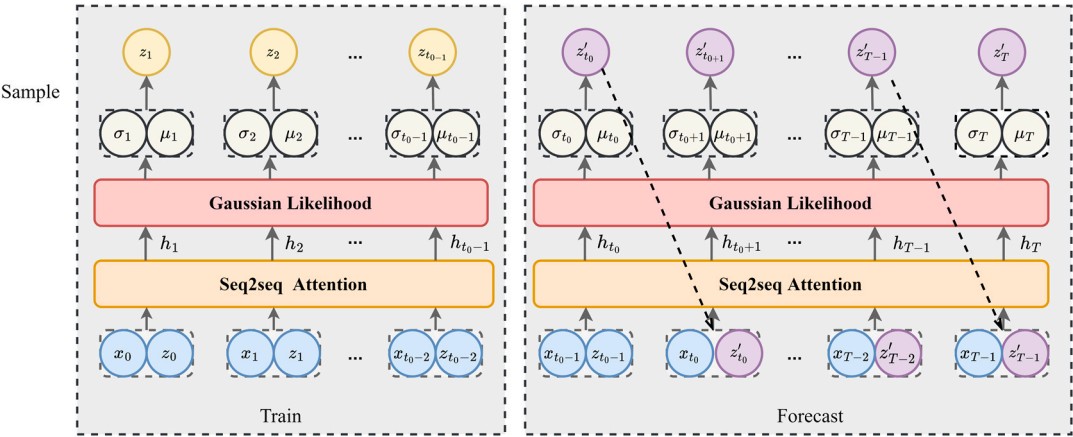

**Figure 5.** Egg production rate prediction model architecture.

During the training phase, at each time point t, the network input includes the previous time point's value $z_{t-1}$, the covariate $x_t$, and the output of the neural network from the previous time step $h_{t-1}$. The model parameters are fitted by maximizing the log-likelihood function $\ell$.

During the prediction phase, the distribution probability p for each time step t is obtained, and the sample value $z_t$ that conforms to p is obtained through Monte Carlo sampling. The median value of the sample value and the covariate $x_{t+1}$ are jointly input into the next time step network, and the process is iterated to obtain the sample interval values for the time range $t_0 : T$.

The specific form of $\theta(ht)$ depends on the likelihood function $\ell(z|\theta)$. In this study, we assume that the egg production rate follows a Gaussian distribution, with the calculation formula as follows:

$$\ell(z|\mu, \sigma) = \frac{1}{\sqrt{2\pi\sigma^2}} e^{\frac{-(z-\mu)^2}{2\sigma^2}} \tag{8}$$

$$\mu(h_t) = wh_t + b \tag{9}$$

$$\sigma(h_t) = \lg(1 + \exp(wh_t + b)) \tag{10}$$

The symbols $\mu$ and $\sigma$ represent the mean and variance. We can analyze and predict the egg production rate with the obtained values of y, $\mu$, and $\sigma$.

### 2.6. Experimental Setup

The experimental environment used in this study is Intel(R) Core(TM) I7-12700H 2.3GHz CPU, NVIDIA GeForce RTX3060 GPU, 16GB RAM, Microsoft Window11 system,

and Anaconda3 IDE with a deep learning framework based on the PyTorch 1.8.1 framework and Python 3.8.5. Furthermore, XGBoost, LightGBM, and CatBoost were utilized via toolkits xgboost 1.6.1, lightgbm 3.3.2, and catboost 1.1, and the feature selection process was conducted using default parameters for the models.

*2.7. Data Acquisition and Preprocessing*

2.7.1. Datasets

This study conducted experiments using Chinese goose breeding tracking data collected by monitoring devices. The raw data consisted of 7 environmental variables, including carbon dioxide, temperature, humidity, dust, light, ammonia, hydrogen sulfide, and egg production data. These environmental variables were sampled every 2 min, resulting in 159,318 data points. Egg production data were collected once daily, resulting in 223 data points.

2.7.2. Address the Issue of Missing Values

During the process of data acquisition, it is inevitable that environmental sensors will experience minor data loss. Therefore, it is necessary to fill in the missing values to ensure the completeness of the data. Since the data collection interval is short and the percentage of missing values is low, accounting for only 0.7% of the total data, we employed the last observation carried forward (LOCF) method to fill in the missing environmental time-series data. As a result, we obtained 160,560 environmental variable data points for our experiment.

2.7.3. Data Conversion and Division

We processed the raw data collected by environmental monitoring equipment. Initially, we averaged the data for seven environmental variables over time intervals of 4 h, 6 h, and 12 h to determine the amount of data required for multistep forecasting. We then generated forecasts with 48 h time steps, using 12, 8, and 4 steps. After conducting multiple experiments, we found that the data volume and prediction step size of the time interval of 6h were moderate compared to those of 4 h and 12 h, and the accuracy of multistep direct prediction was higher compared to the other two time intervals. Therefore, we averaged the data over a 6 h time interval, resulting in 892 data points for multistep prediction. We further calculated the daily average of the environmental data, resulting in 223 data points that were used as covariates for egg production rate prediction. Additionally, we converted the egg production data to egg production rate data using the following formula:

$$Egg\ production\ rate = \frac{Daily\ egg\ production}{Number\ of\ waterfowls\ per\ day} \tag{11}$$

The data were transformed into supervised learning data. Historical egg production rate data and environmental variables were used as inputs, which output the prediction interval of the egg production rate for the upcoming three days recursively. The median value was considered as the point prediction result. All experiments were conducted using 60% training data and 40% testing data.

2.7.4. Normalization

As shown in Table 2 and Figure 6, there are significant differences in the dimensions between different data, which can weaken the predictive accuracy of the model's convergence speed. This study adopts the Min–Max normalization method to map the data to the range of [0, 1], with the following formula:

$$X_{new} = \frac{X - X_{min}}{X_{max} - X_{min}} \tag{12}$$

**Table 2.** The original data with a time interval of 24 h.

| Date | Egg Production Rate | Light (lx) | Dust (ppm) | Carbon Dioxide (ppm) | Temperature (°C) | Humidity (%) | Ammonia (ppm) | Hydrogen Sulfide (ppm) |
|---|---|---|---|---|---|---|---|---|
| 11 October 2018 | 0.9884 | 335.77 | 65.69 | 467.86 | 29.07 | 98.02 | 3.56 | 1 |
| 12 October 2018 | 0.9884 | 354.46 | 53.20 | 470.21 | 30.16 | 95.09 | 3.79 | 1 |
| 13 October 2018 | 0.9884 | 430.51 | 42.87 | 450.72 | 29.28 | 96.58 | 3.93 | 1 |
| 26 March 2019 | 0.9487 | 251.93 | 48.64 | 497.11 | 24.84 | 99.01 | 2.08 | 1 |
| 27 March 2019 | 0.9487 | 133.46 | 38.65 | 484.52 | 25.01 | 100.00 | 1.96 | 1 |
| 28 March 2019 | 0.9487 | 133.46 | 37.99 | 490.72 | 24.81 | 100.00 | 2.09 | 1 |

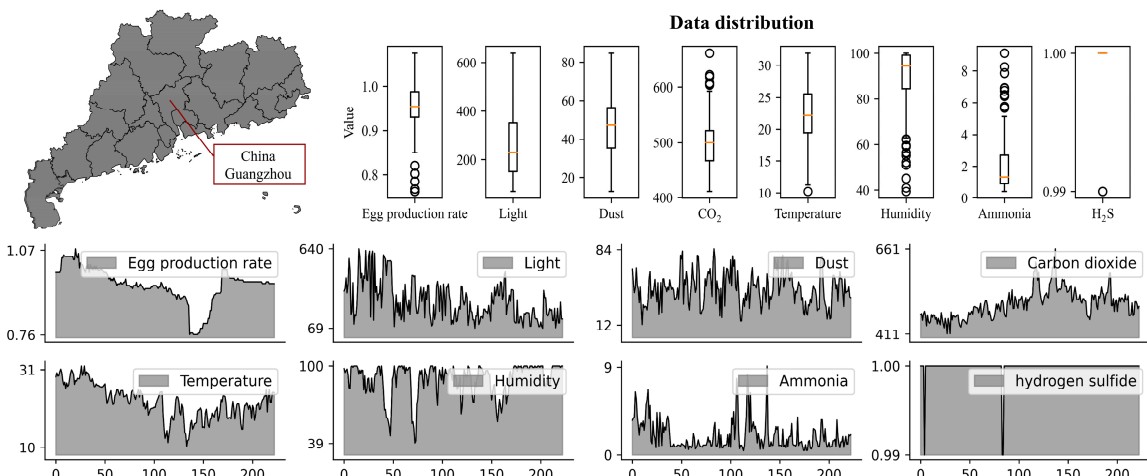

**Figure 6.** The information of the dataset.

## 2.8. Evaluation Metrics

### 2.8.1. Point Prediction Evaluation Metrics

To evaluate the point prediction performance of the model, root mean square error (RMSE), mean absolute error (MAE), and mean absolute percentage error (MAPE) were employed as evaluation metrics to measure the deviation between the actual egg production rate and expected egg production rate. The formulas for the selected evaluation metrics are as follows:

$$RMSE = \sqrt{\frac{1}{n}\sum_{i=1}^{n}(Y_i - y_i)^2} \tag{13}$$

$$MAE = \frac{1}{n}\sum_{i=1}^{n}|Y_i - y_i| \tag{14}$$

$$MAPE = \frac{100\%}{n}\sum_{i=1}^{n}\left|\frac{Y_i - y_i}{y_i}\right| \tag{15}$$

where $n$ represents the number of predicted samples; $Y_i$ represents the prediction result; $y_i$ represents the actual value.

### 2.8.2. Interval Prediction Evaluation Metrics

The interval prediction performance is evaluated based on prediction interval coverage probability (PICP), prediction interval normalized root mean square width (PINRW), and coverage width-based criterion (CWC). The PICP metric indicates the probability of actual values falling within the predicted interval. A higher PICP indicates greater actual values falling within the interval and a higher reliability of interval prediction. The PINAW and PINRW metrics are used to measure the width of the interval, mainly to prevent the model from pursuing reliability at the expense of the width of the interval, making the

predicted values unable to effectively describe uncertainty information. The CWC metric is a comprehensive evaluation metric considering prediction reliability and interval width.

$$PICP = \frac{1}{N}\left(\sum_{i=1}^{N} C_i\right)$$
$$Here, C_i = \begin{cases} 1, & y_i \in \left[y_i^U, y_i^L\right] \\ 0, & y_i \notin \left[y_i^U, y_i^L\right] \end{cases} \tag{16}$$

$$PINAW = \frac{1}{NA}\sum_{i=1}^{N}\left|y_i^U - y_i^L\right| \tag{17}$$

$$PINRW = \frac{1}{A}\sqrt{\frac{1}{N}\sum_{i=1}^{N}(y_i^U - y_i^L)^2} \tag{18}$$

$$CWC = PINAW\left(1 + \gamma e^{-\eta(PICP-\mu)}\right)$$
$$Here, \gamma = \begin{cases} 0, & PICP \geq \mu \\ 1, & PICP < \mu \end{cases} \tag{19}$$

where $y_i^U$ represents the predicted upper bound while $y_i^L$ indicates the predicted lower bound, $y_i$ stands for the actual value, and A denotes the target value range used for data normalization. The parameters $\gamma$, $\eta$ and $\mu$ are hyperparameters that measure the CWC metric. The value of $\mu$ determines the minimum acceptable parameter for the PICP metric, which is set at 0.95. In this case, the parameter $\eta$ is the penalty factor for unacceptable PICP and is set at 1. Therefore, when the PICP metric is not met, it significantly impacts the CWC metric more. In contrast, PINAW has a more significant impact on the CWC metric in the opposite case.

## 3. Results

### 3.1. Feature Selection of Environmental Variables

The experiment utilized environmental data with a time interval of 24 h and employed three models, XGBoost, LightGBM, and CatBoost, along with the RFE method for feature selection. Firstly, to investigate the effect of different environmental variables on the egg production rate, we used environmental variables and the egg production rate as the inputs and outputs of the XGBoost, LightGBM and CatBoost models. Based on the feature importance calculation function of the weak learner decision trees, we can obtain the importance ranking of environmental variables as shown in Figure 7a and Table 3. From the feature importance graph, it is evident that the feature importance ranking in descending order is as follows: carbon dioxide, temperature, humidity, dust, noise, light intensity, and hydrogen sulfide for the XGBoost and CatBoost models. However, for the LightGBM model, the importance of temperature is higher than that of carbon dioxide, and the ranking of other variables is the same as the other two models. In addition, we observed that the impact of hydrogen sulfide on the egg production rate could be almost ignored among the seven environmental variables.

**Table 3.** Importance score of environmental variables.

| Variable | XGBoost | LightGBM | CatBoost |
|---|---|---|---|
| Carbon dioxide | 0.533 | 0.195 | 0.400 |
| Temp | 0.225 | 0.256 | 0.250 |
| Humidity | 0.207 | 0.162 | 0.169 |
| Dust | 0.017 | 0.155 | 0.066 |
| Ammonia | 0.011 | 0.126 | 0.052 |
| Light | 0.007 | 0.105 | 0.062 |
| Hydrogen sulfide | $4.16 \times 10^{-5}$ | 0.0 | 0.002 |

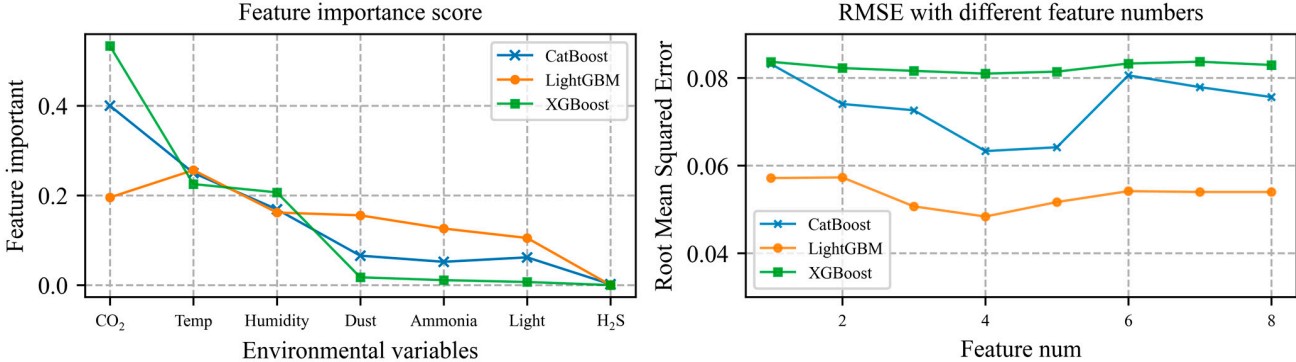

(**a**) Importance score of environment variables.  (**b**) Prediction performance of different feature subsets.

**Figure 7.** Distribution of importance score for environmental variables and prediction performance of different feature subsets.

We transformed egg production rate data into supervised learning data to further explore the optimal number of features for inputting into the model. Combining RFE with seven environmental variables and the egg production rate from the previous period (eight features), we gradually filtered out the low-importance features according to each model's feature importance sequence. We used the root mean square error as the evaluation metric, and observed the fitting effect of the model under different feature numbers. As shown in Figure 7b, we found that when the feature number was four (the egg production rate from the previous time, carbon dioxide, temperature, and humidity), the RMSE of the three models reached the lowest point, indicating that the three models achieved the best prediction performance. Therefore, we chose carbon dioxide, temperature, humidity, and the egg production rate from the previous time as the input for the final model. In Section 3.4.3, we will provide additional evidence to demonstrate the effectiveness of this feature selection method.

### 3.2. GWO-VMD Decomposition of Critical Environmental Variables

The time series of the environmental variables exhibited high complexity and significant volatility, indicating a strong mutual interference among the sequences of different frequencies within the environmental variables. To enhance the accuracy and stability of the multistep direct prediction of environmental variables, we adopted the GWO-VMD method. We utilized the minimum envelope entropy value as an adaptive parameter to decompose each complex environmental variable into multiple sub-series of varying frequencies.

According to the feature selection results, the experiment employed time-series data of carbon dioxide, temperature, and humidity with a time interval of 6 h as the data sample. The initial and maximum numbers of iterations for the GWO algorithm were set to 20. Since it was necessary to optimize two parameters, namely K and $\alpha$, the variable dimension is set to 2. The value range of K was set to [2, 9], and $\alpha$ was set to [400, 3000]. The optimization process of the three environmental variables using the GWO-VMD method is shown in Figure 8. It can be observed that the GWO method shows excellent optimization ability and fast convergence speed in optimizing VMD parameters. The total number of decomposition modes, quadratic penalty coefficient, and decomposition results of the three environmental variables are shown in Table 4 and Figure A1 (in Appendix A). After VMD decomposition, the intrinsic mode functions (IMFs) of different frequencies can be evenly displayed in the time domain. This indicates that VMD decomposition could effectively reduce the non-linearity, non-stationarity, and randomness of complex environmental variables, and fully extract the time-series characteristics of environmental variables.

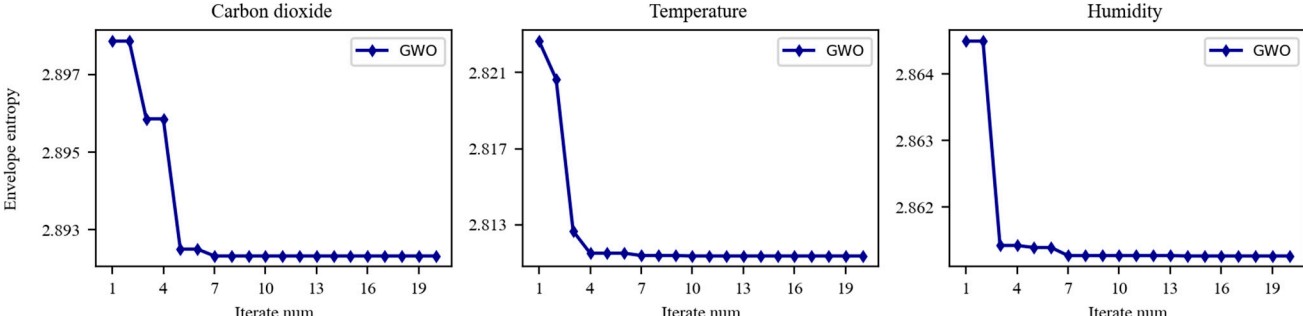

**Figure 8.** GWO-VMD decomposition curves for important environmental variables.

**Table 4.** VMD decomposition parameters for important environmental variables.

| Variable | Envelope Entropy | K | $\alpha$ |
|---|---|---|---|
| Carbon dioxide | 2.8923 | 2897.673 | 7 |
| Temperature | 2.8113 | 3012.798 | 5 |
| Humidity | 2.8612 | 1795.127 | 4 |

### 3.3. LSTM-Based Multistep Prediction of Environmental Variables

The time scale of the data entered into the egg production rate prediction model is 24 h. However, we have processed the environmental variable time interval into 6 h. Therefore, the output steps for multistep prediction are 4 (24 h) and 8 (48 h), and we can obtain the environmental variable data for the next two days by averaging the prediction results daily. To achieve better multistep prediction results, we transformed the IMFs of the three environmental variables, which have undergone GWO-VMD decomposition, into supervised learning data. For this purpose, we needed to set the sliding window length. Different sliding window lengths can affect the prediction performance of the LSTM model. A window that is too short or too long will weaken the simulation effect of the LSTM and reduce the prediction accuracy. This experiment set the LSTM input sequence's time window lengths to 8 and 16 and used the previous 8 and 16 time periods to predict the following 4 and 8 time periods, respectively. The prediction results were then reconstructed and averaged daily, and the visualized results are shown in Figure 9.

To demonstrate the necessity of the proposed approach, direct multistep prediction was performed on the environmental variable data with time intervals of 6 h and 24 h. The results were compared using the mean absolute error (MAE) as the evaluation metric, as shown in Table 5. We observed that predictions for data with a time interval of 6 h and then averaging the results, the MAE was lower than predictions using data with a time interval of 24 h. This suggests that reducing the time scale and increasing the amount of data can effectively improve the multistep prediction accuracy of environmental time series. In addition, after GWO-VMD decomposition, the multistep prediction performance was significantly improved, indicating that the VMD method can effectively reduce the complexity of the time series and remove noise.

**Table 5.** Comparison of MAE prediction errors for important environmental variables under different processing methods.

| Environment Variable | 6 h Average and GWO-VMD | | 6 h Average | | 24 h Average | |
|---|---|---|---|---|---|---|
| | 1 Day | 2 Days | 1 Day | 2 Days | 1 Day | 2 Days |
| Carbon dioxide | 7.987 | 11.100 | 15.710 | 17.930 | 18.048 | 23.758 |
| Temperature | 0.800 | 0.964 | 1.434 | 1.540 | 1.421 | 1.989 |
| Humidity | 1.670 | 2.087 | 2.200 | 4.306 | 3.858 | 4.705 |

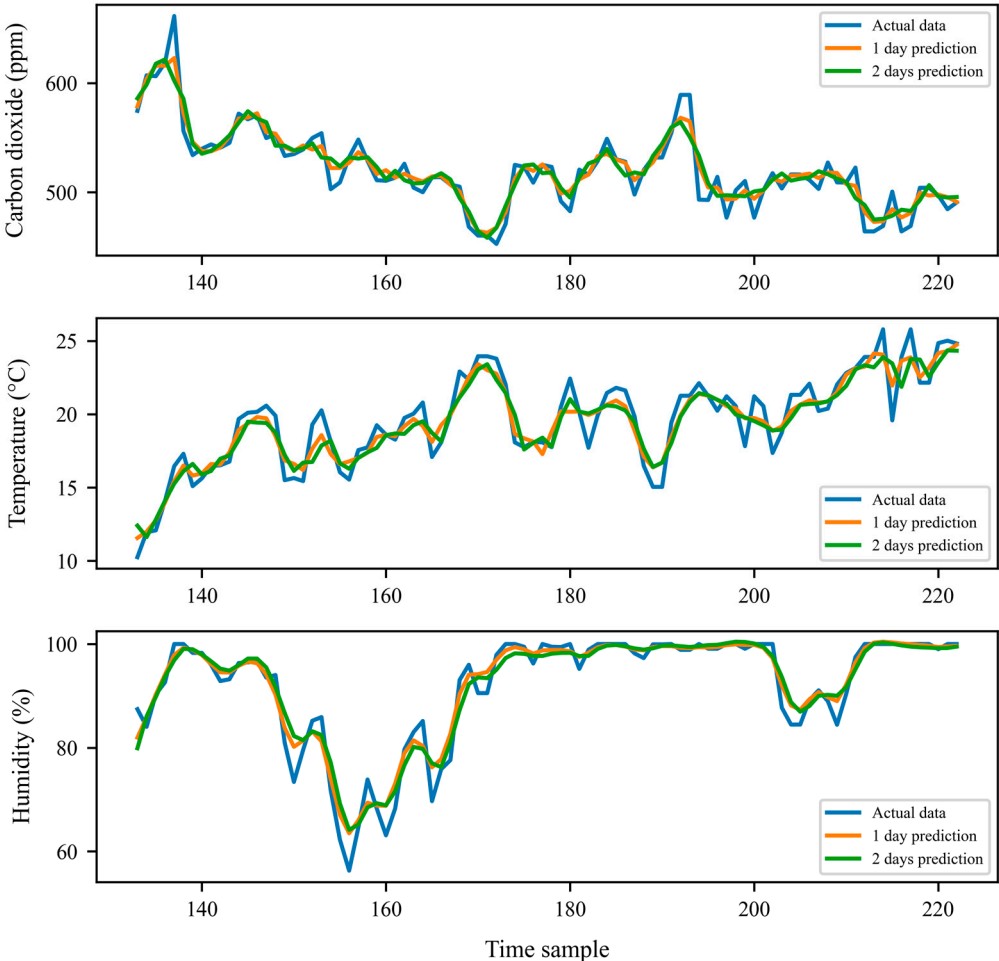

**Figure 9.** Prediction results for important environmental variables.

### 3.4. Prediction of Egg Production Rate

After obtaining important future environmental variables (carbon dioxide, temperature, humidity) for a time interval of 24 h, they were entered into the final model together with the egg production rate of the previous period for point and interval prediction. For the sake of discussion, we abbreviate the model proposed in this study as SAG (Seq2seq-Attention-Gaussian). Meanwhile, we use the multi-objective optimization algorithm MOGWO to optimize the objective functions MAE and CWC, and we abbreviate the model combined with the multi-objective optimization algorithm as MSAG (MOGWO-Seq2seq-Attention-Gaussian).

#### 3.4.1. Point Prediction

In this experiment, we compared the performance of the MSAG model with several commonly used machine learning and deep learning models, including multilayer perceptron (MLP), random forest (RF), least squares support vector machine (LSSVM), long short-term memory (LSTM), gated recurrent unit (GRU), deep autoregressive (DeepAR), mixed-quantile recurrent neural network (MQRNN), and SAG. All models are trained and evaluated using the same input variables as MSAG. The comparative test results are shown in Table 6 and Figure 10.

**Table 6.** Comparison results of different models for point prediction.

| Model | 1-Step | | | 2-Step | | | 3-Step | | |
|---|---|---|---|---|---|---|---|---|---|
| | **MAE** | **MAPE** | **RMSE** | **MAE** | **MAPE** | **RMSE** | **MAE** | **MAPE** | **RMSE** |
| MSAG | 0.0053 | 0.5985 | 0.0089 | 0.0062 | 0.7057 | 0.0132 | 0.0068 | 0.7677 | 0.0127 |
| SAG | 0.0059 | 0.6640 | 0.0100 | 0.0069 | 0.7804 | 0.0139 | 0.0081 | 0.9210 | 0.0151 |
| DeepAR | 0.0066 | 0.7542 | 0.0130 | 0.0106 | 1.2019 | 0.0177 | 0.0111 | 1.2516 | 0.0175 |
| MQRNN | 0.0078 | 0.8494 | 0.0094 | 0.0111 | 1.2249 | 0.0133 | 0.0187 | 2.0598 | 0.0210 |
| GRU | 0.0075 | 0.8657 | 0.0156 | 0.0121 | 1.3822 | 0.0200 | 0.0172 | 1.9669 | 0.0265 |
| LSTM | 0.0079 | 0.9025 | 0.0157 | 0.0114 | 1.2994 | 0.0198 | 0.0137 | 1.5569 | 0.0245 |
| LSSVM | 0.0155 | 1.7488 | 0.0188 | 0.0190 | 2.1188 | 0.0222 | 0.0199 | 2.2082 | 0.0239 |
| RF | 0.0137 | 1.6159 | 0.0234 | 0.0168 | 1.9668 | 0.0271 | 0.0315 | 3.4891 | 0.0412 |
| MLP | 0.0262 | 2.9749 | 0.0327 | 0.0315 | 3.5526 | 0.0397 | 0.0359 | 4.0450 | 0.0438 |

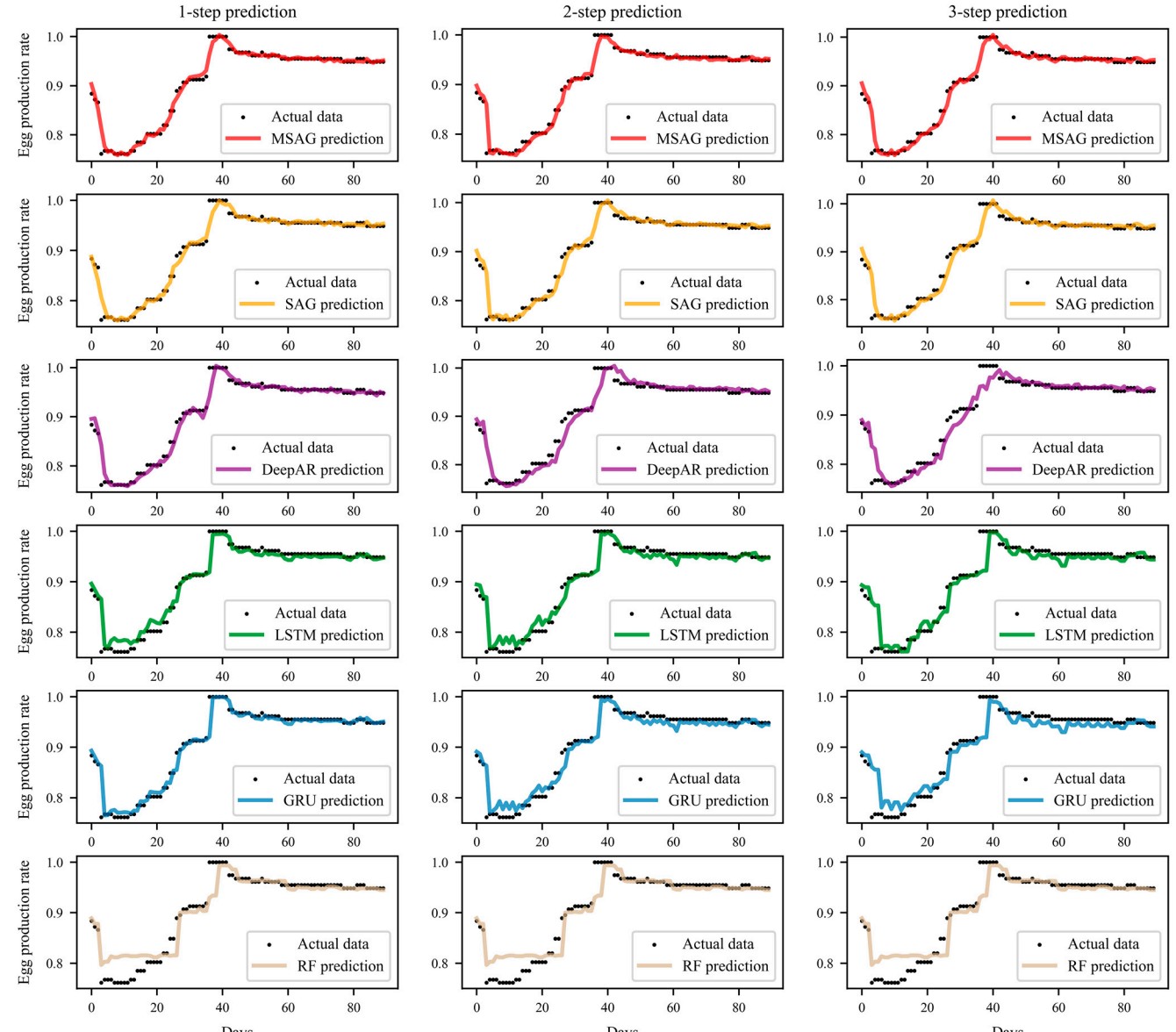

**Figure 10.** Comparison of different models for point prediction.

3.4.2. Interval Prediction

To validate the effectiveness of the proposed interval prediction model, we compared the MSAG model with the interval prediction models DeepAR, MQRNN and SAG. A

confidence interval of 10% to 90% was chosen, and the results of the comparative tests are presented in Table 7 and Figure 11.

**Table 7.** Comparison results of different models for interval prediction.

| Model | 1-Step | | | 2-Step | | | 3-Step | | |
|---|---|---|---|---|---|---|---|---|---|
| | PICP | PINRW | CWC | PICP | PINRW | CWC | PICP | PINRW | CWC |
| MSAG | 1.0000 | 0.1886 | 0.1688 | 0.9778 | 0.1793 | 0.1735 | 0.9778 | 0.2108 | 0.2076 |
| SAG | 0.9888 | 0.2068 | 0.1890 | 0.9667 | 0.1893 | 0.1812 | 0.9667 | 0.2132 | 0.2099 |
| DeepAR | 0.9667 | 0.1985 | 0.1940 | 0.9778 | 0.2085 | 0.1831 | 0.9888 | 0.2588 | 0.2348 |
| MQRNN | 0.8666 | 0.1417 | 0.2572 | 0.8333 | 0.1995 | 0.3649 | 0.8000 | 0.2139 | 0.3980 |

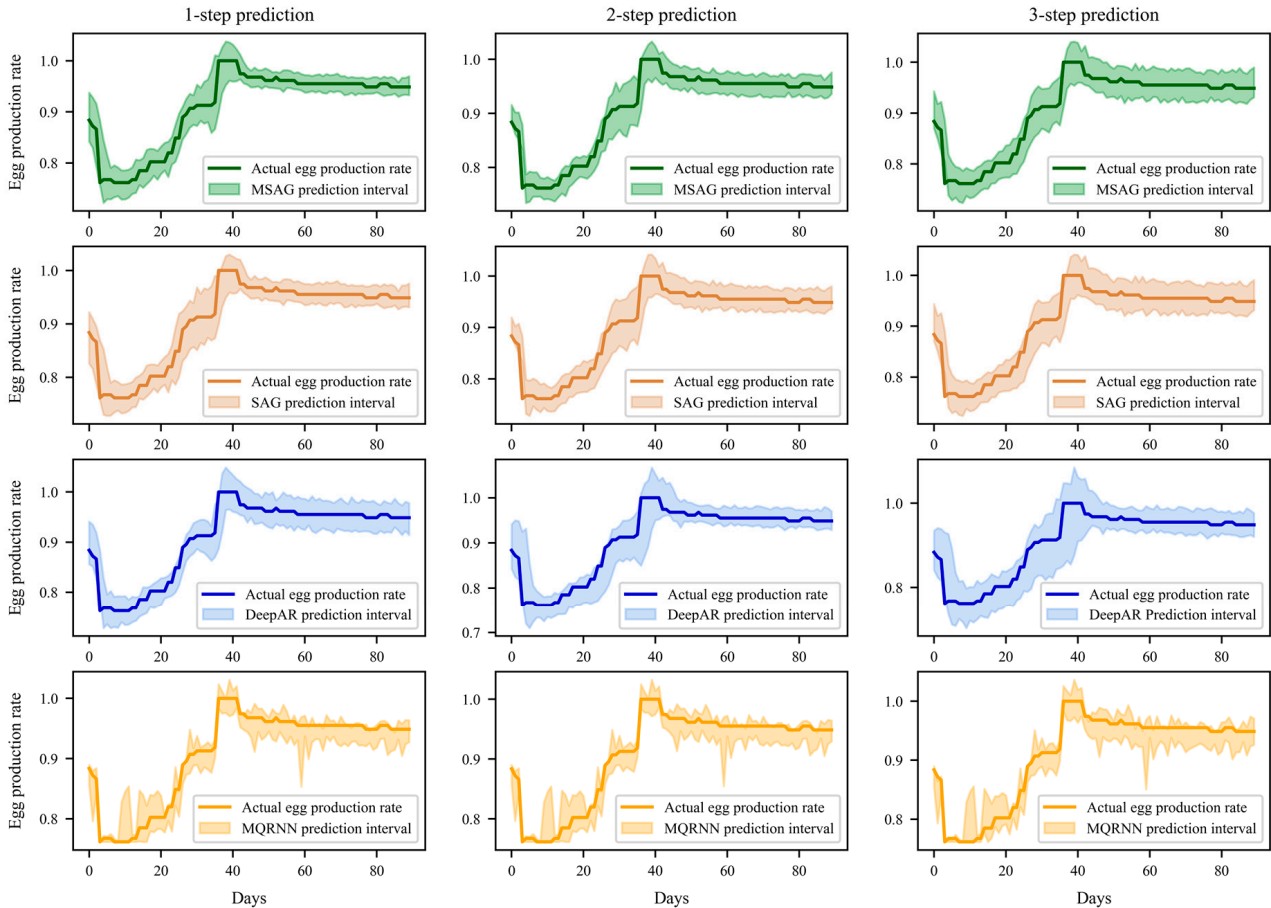

**Figure 11.** Comparison of different models for interval prediction.

### 3.4.3. Prediction Using Different Feature Sets

To validate the effectiveness of the feature selection method, we utilized the feature importance ranking based on RFE, and selected different feature subsets to input into the MSAG model for single-step point and interval prediction. The comparative test results are shown in Table 8, from which we can observe that MSAG-4 performs the best overall, while MSAG-1 has only historical egg production rate data as the input and does not incorporate environmental variables, so its prediction effect is the worst.

**Table 8.** Comparison results of different feature numbers.

| Metrics | MSAG-1 | MSAG-2 | MSAG-3 | MSAG-4 | MSAG-5 | MSAG-6 | MSAG-7 | MSAG-8 |
|---------|--------|--------|--------|--------|--------|--------|--------|--------|
| MAE | 0.0143 | 0.0076 | 0.0067 | 0.0053 | 0.0062 | 0.0062 | 0.0069 | 0.0066 |
| RMSE | 0.0204 | 0.0115 | 0.0118 | 0.0089 | 0.0112 | 0.0101 | 0.0134 | 0.0113 |
| MAPE | 1.6029 | 0.8552 | 0.7654 | 0.5985 | 0.7066 | 0.6948 | 0.7923 | 0.7540 |
| PICP | 0.9333 | 1.0000 | 0.9778 | 1.0000 | 0.9889 | 0.9889 | 0.9889 | 1.0000 |
| PINRW | 0.3278 | 0.2546 | 0.2023 | 0.1886 | 0.1991 | 0.1995 | 0.2005 | 0.2304 |
| CWC | 1.1868 | 0.2335 | 0.1928 | 0.1688 | 0.1711 | 0.1706 | 0.1715 | 0.1941 |

## 4. Discussion

Various environmental factors have a certain degree of impact on the egg-laying performance of poultry. However, a single environmental factor cannot determine egg production. Therefore, it is necessary to consider multiple factors comprehensively. This study used XGBoost, LightGBM, and CatBoost models, along with the recursive feature elimination (RFE) method to screen seven environmental variables. By examining the feature importance ranking chart, it can be observed that carbon dioxide, temperature, and humidity within the goose house significantly impact egg production performance more than the other environmental variables. Additionally, Table 8 shows that the best point prediction and interval prediction results were achieved when using four features. Compared to no feature selection, the mean absolute error (MAE) of the model was reduced by approximately 19.7%, demonstrating that this method is scientifically effective for data dimensionality reduction. This method can serve as scenario analysis to analyze environmental variables, reduce input dimensions and data training time, and improve prediction accuracy, which has practical significance.

The accuracy of predicting future environmental variables as covariates is also critical for predicting egg production rates based on recursive multistep prediction methods. Reducing the temporal scale of data, increasing the amount of data, and decomposing the data using the GWO-VMD method can increase the predictive accuracy of environmental variables. The method is significantly better than direct prediction, with MAE reductions of about 53.3%, 51.5% and 55.6% for the three important environmental variables predicted. To address the parameter optimization problem in the VMD decomposition process, a grey wolf optimization algorithm with the local minimum envelope entropy as the objective function was used to achieve adaptive optimization of K and $\alpha$. This method effectively decomposes the features of different frequencies in environmental variables and reduces the influence of noise, which has good application prospects.

This study proposed a multistep point and interval prediction model for the egg production rate based on the Gaussian distribution, seq2seq, and attention mechanism. In addition, the multi-objective optimization algorithm MOGWO searched the initialization parameters to ensure the effectiveness of point and interval predictions. In terms of point prediction, as shown in Figure 12, we extracted four time periods with large fluctuations in the egg production curves and overall prediction deviations. Compared with other models, the prediction curve of MSAG can still closely follow the trend of egg production rate data in areas with large fluctuations or transitions, and the fit is better. With increasing prediction steps, the prediction curve offset is smaller, showing excellent stability and accuracy. Compared with the worst-performing MLP model, the MSAG model predicts a reduction in MAE of about 79.8%, 80.3%, and 81.0% for the next three time periods respectively. In terms of interval prediction, MSAG also performs well. Compared to the DeepAR model, although the MSAG model has a lower PICP index when the prediction time steps are 3, the PINRW index of MSAG is much lower than DeepAR. The predicted interval is narrower with a higher CWC index, indicating that MSAG's predicted interval can better describe the uncertain information of the egg production rate. Similarly, when the prediction time step is 1, although the PINRW index of MQRNN is lower than that of MSAG, the PICP is much lower than the penalty boundary of 0.95, indicating that MQRNN

cannot describe the change information of the egg production rate well. Therefore, the MSAG model has higher stability in interval prediction than the other two models and can form appropriate prediction intervals.

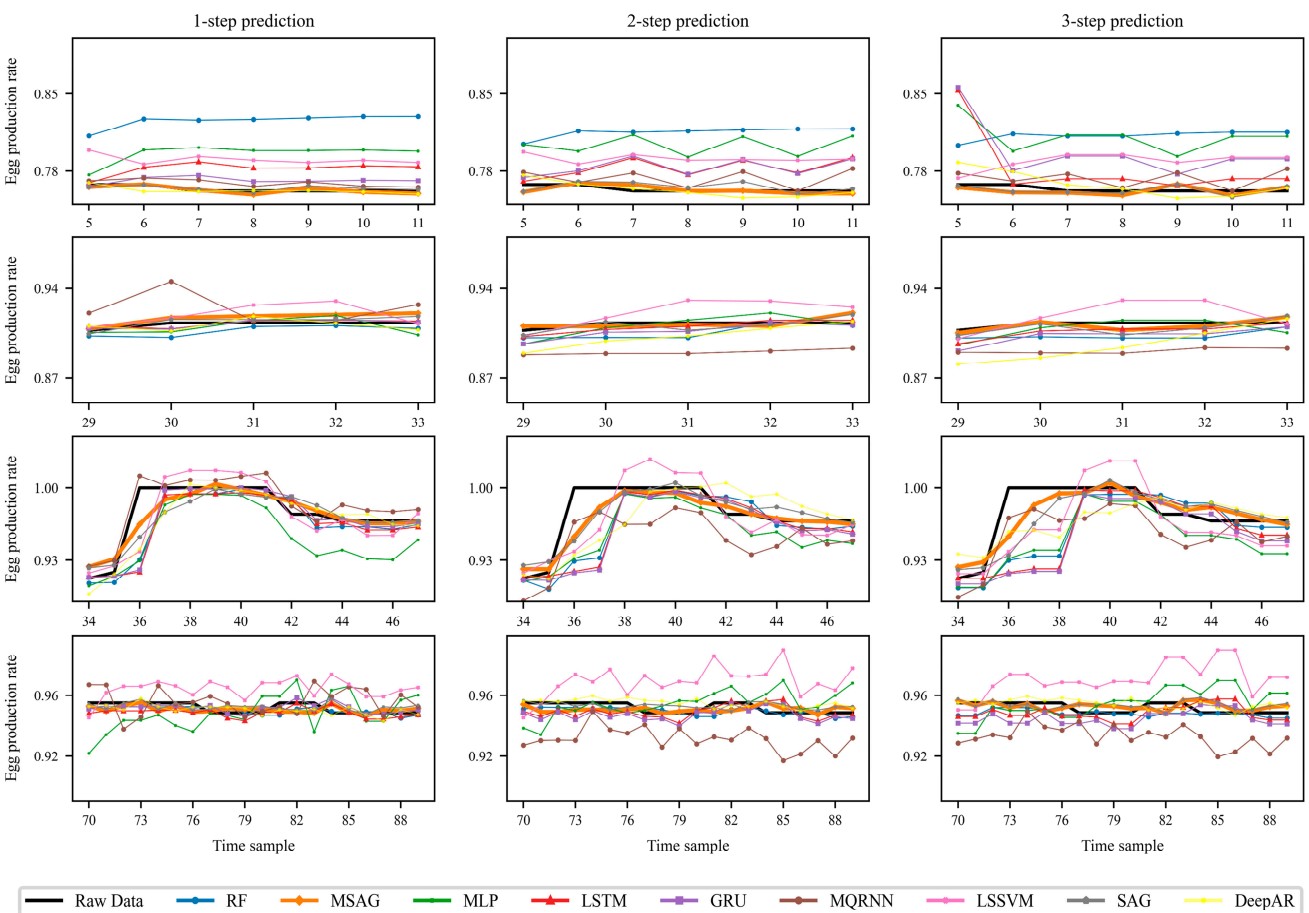

**Figure 12.** Comparison of different models for predicting local areas with larger deviations.

The results in Tables 6 and 7 show that SAG has good point prediction and interval prediction performance compared to other models. However, MSAG optimized by MOGWO parameters has superior performance compared to SAG, with MAE reductions of 10.1%, 10.1%, and 16.0% for predicting the next 1–3 steps, indicating that MOGWO helps us to search for model parameter initialization, reduce parameter tuning time and improve prediction performance.

Overall, through Table 6 and Figure 12, we can observe that as the forecasting time steps increase, the cumulative error leads to a rapid decline in the predictive performance of many models, resulting in significant deviations in places with high volatility. For instance, from step 1 to step 3, the MAE of LSTM and DeepAR increased by 42.3% and 40.5%, respectively. However, the MSAG model exhibited a smaller increase rate of only 22.0% in MAE compared to other models. This is because we incorporated an attention mechanism on top of the seq2seq structure, which can better capture the correlation between the past and future time series and reduce the loss of useful information. Similarly, the stable performance of MSAG in interval prediction is also related to the attention mechanism.

## 5. Conclusions

The egg production rate is a non-linear time series influenced by various environmental factors. This study proposes a novel recursive multistep prediction method for the waterfowl egg production rate. Firstly, the XGBoost, LightGBM, and CatBoost models were combined with the RFE feature reduction idea to screen the environmental variables that

affect the egg production rate, reducing the dimensionality of the input model. Then, based on the feature selection results, the time scale was reduced and the GWO-VMD method was used for decomposition to increase the amount of data for the environmental variables and reduce their complexity. The LSTM neural network was used for direct multistep prediction and daily averages to obtain the environmental variables for the next two days. Finally, the environmental variables and the egg production rate from the previous time were input into the proposed model for point and interval predictions for the next three days. The experimental results show that combining critical environmental variables for recursive multistep prediction effectively predicts the egg production rate. The experimental results show that the feature selection method employed in this study significantly reduces the training time and improves prediction accuracy. Compared to other existing egg production rate prediction models, the proposed model, which combines the seq2seq structure and attention mechanism, can effectively prevent the problem of decreasing prediction accuracy due to the increased prediction steps. Compared with the MLP model, the MSAG model predicts a reduction in MAE of about 79.8%, 80.3% and 81.0% for the next three time periods, respectively, showing good prediction performance.

Additionally, this study introduces interval prediction into the field of egg production rate prediction for the first time. The prediction interval obtained from the proposed model can cover the egg production rate curve well with a narrow coverage interval, providing more helpful information for livestock breeders. This study provides guidance for estimating egg production rates of livestock and poultry from an environmental perspective, which is of great significance for realizing intelligent breeding and improving livestock and poultry production efficiency.

In the future, we will continue our research mainly in the following two areas: (1) analyzing and incorporating more factors related to egg production, such as diet, water intake, and body weight, to build a more comprehensive and efficient egg production rate prediction framework; (2) exploring more effective methods for predicting egg production intervals to achieve better prediction results.

**Author Contributions:** Conception, H.Y. and Z.W.; methodology, H.Y., Z.W. and J.-C.W.; software, Z.W. and Y.C.; validation, H.Y., Z.W., J.-C.W., Y.C. and M.C.; formal analysis, H.Y., M.C. and S.L.; investigation, H.Y., Z.W. and L.G.; resource, H.Y. and S.G.H.; data, H.Y. and Z.W.; writing—original draft preparation, H.Y., Z.W. and S.G.H.; writing—review and editing, H.Y., Z.W. and L.G.; projection administration, H.Y. All authors have read and agreed to the published version of the manuscript.

**Funding:** The work was supported by the National Natural Science Foundation of China (61871475); Guangdong Natural Science Foundation (2021A1515011605); Opening Foundation of Xinjiang Production and Construction Corps Key Laboratory of Modem Agricultural Machinery (BTNJ2021002); Guangzhou Innovation Platform Construction Project (201905010006); Guangdong Province Science and Technology Plan Project (2019B020215003); Yunfu Science and Technology Plan Project (2022020302) and Key R & D projects of Guangzhou (202103000033).

**Institutional Review Board Statement:** Not applicable.

**Data Availability Statement:** The data presented in this study are available upon request from the corresponding author.

**Conflicts of Interest:** The authors declare no conflict of interest.

**Appendix A**

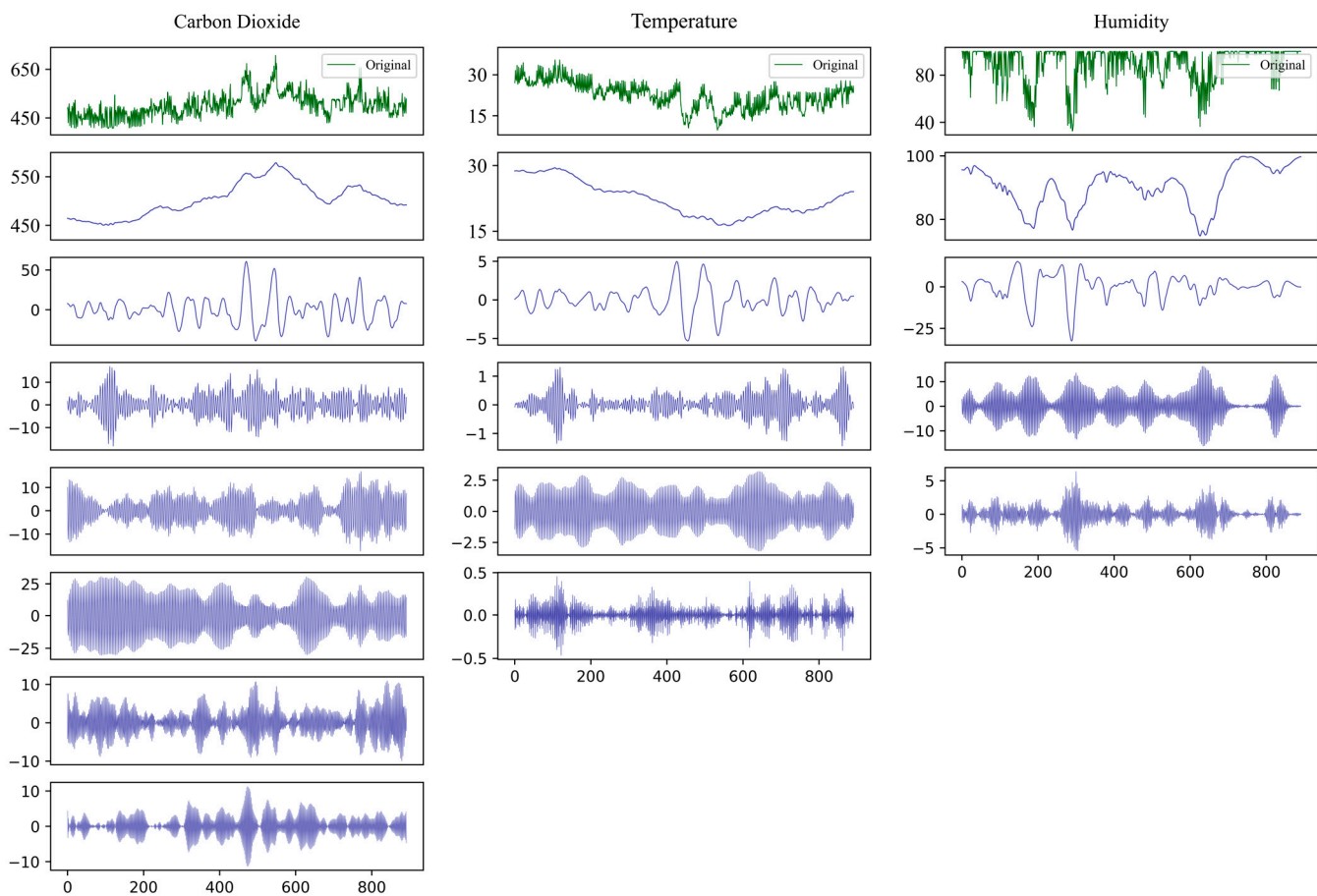

**Figure A1.** VMD decomposition results of importance environmental variables.

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
