# Peer review of "A Multistep Interval Prediction Method Combining Environmental Variables and Attention Mechanism for Egg Production Rate"

_agriculture, doi:10.3390/agriculture13061255_

Round 1

Reviewer 1 Report

 A novel recursive multistep prediction method for waterfowl egg production rate is presented in the paper.

The environmental variables prediction and the relation with egg production prediction might be discussion with more details. It is not very clear which type of units are used for environment parameters. How about the measurement accuracy of environment data, particularly CO2. More details about data collection may be considered.

Please check and include the SI units associated with tables and graph axes.

A comparison with existing methods in the field might be highlighted. Please consider additional discussion for Figure 13. 

The originality of the work might be clearly presented in the conclusion part.

The English usage might be verified for the whole paper.

Reviewer 2 Report

Recommendations to authors

General comments: 

Well written manuscript providing all the necessary information. Some suggestions are given below for further improvement.

More detailed comments:

- Line 377: Replace "Missing values repair" with "Address the issue of missing values".

- Line 381: Replace "small" with "low".

- Line 390: Please improve "… we found that the amount of data and the number of prediction steps were moderate …".

- Line 397: Maybe "waterfowls" instead of "waterfowl"?

- Line 403: Although beautiful and easy to comprehend, I think the color is just "noise" here. Please consider using only black or grey for the diagrams with captions a bit bigger to improve readability. The part of the figure 7 that corresponds to "Data distribution" must be replaced with boxplots. Numbers in "Basic Information" must be black (instead of blue). In fact, the part that shows the ‘Basic Information’ can be removed from the figure 7 since it is being presented in Table 1.

- Line 405: Bold in ‘Dates’ and ‘column titles’ is not necessary.

- Line 446: You must explain or elaborate on how the importance scores/rankings have been obtained.

- Line 470: Replace "Feature name" with "Environmental variables".

- Line 513: Make the correction: replace "Actaul” with “Actual”.

- Line 546, Figure 11: The graphs are very complex and “overcrowded”. Consider grouping models and provide more graphs (for each step forecast) to improve readability. In its current form, these three graphs do not offer much to the understanding of the data being presented.

- Line 595: Although this is an important conclusion, it is not obvious that "Compared to other models, the prediction curve of MSAG fits the actual values better" in Figure 13. You must find a way to improve Figure 13 or at least find a better way to show that MSAG is better than the other models. 

- Line 627: A short overview on the quantitative results is missing in conclusions. I think you can add some information about the improvement that current method offers in numbers (i.e., lines 610 – 620).
